# Intranasal oxygen reverses hypoxaemia in immobilised free-ranging capybaras (*Hydrochoerus hydrochaeris*)

**Jefferson F. Cordeiro**[1], **Mariana C. Sanches**[1], **Elidiane Rusch**[1], **Nathalia V. Xavier**[1], **Ana Angélica Cassoli**[1], **Åsa Fahlman**[2], **Adriano B. Carregaro**[1] *

**1** Veterinary Medicine Department, Faculty of Animal Science and Food Engineering, University of São Paulo (USP), Pirassununga, SP, Brazil, **2** Swedish Biodiversity Centre, Department of Urban and Rural Development, Swedish University of Agricultural Sciences (SLU), Uppsala, Sweden

* carregaro@usp.br

**Data Availability Statement:** The data underlying the results presented in the study are available at www.ebi.ac.uk/biostudies/studies/S-BSST655.

## Abstract

Capybara (*Hydrochoerus hydrochaeris*) is the main host of tick-borne pathogens causing Brazilian spotted fever; therefore, controlling its population is essential, and this may require chemical restraint. We assessed the impact of chemical restraint protocols on the partial pressure of arterial oxygen ($PaO_2$) and other blood variables in 36 capybaras and the effect of different flows of nasal oxygen ($O_2$) supplementation. The capybaras were hand-injected with dexmedetomidine (5 μg/kg) and midazolam (0.1 mg/kg) and butorphanol (0.2 mg/kg) (DMB, n = 18) or methadone (0.1 mg/kg) (DMM, n = 18). One-third of the animals were maintained in ambient air throughout the procedure, and one-third were administered intranasal 2 L/min $O_2$ after 30 min whereas the other third were administered 5 L/min $O_2$. Arterial blood gases, acid-base status, and electrolytes were assessed 30 and 60 min after drug injection. The DMB and DMM groups did not vary based on any of the evaluated variables. All animals developed hypoxaemia ($PaO_2$ 44 [30; 73] mmHg, $SaO_2$ 81 [62; 93] %) 30 min before $O_2$ supplementation. Intranasal $O_2$ at 2 L/min improved $PaO_2$ (63 [49; 97] mmHg and $SaO_2$ [92 85; 98] %), but 9 of 12 capybaras remained hypoxaemic. A higher $O_2$ flow of 5 L/min was efficient in treating hypoxaemia ($PaO_2$ 188 [146; 414] mmHg, $SaO_2$ 100 [99; 100] %) in all the 12 animals that received it. Both drug protocols induced hypoxaemia, which could be treated with intranasal oxygen supplementation.

## Introduction

The capybara (*Hydrochoerus hydrochaeris*) is the world's largest rodent, and its population can grow exponentially with a large food supply and in the absence of natural predators [1]. In Brazil, the capybara is an important host for the transmission cycle of Brazilian spotted fever [2]. Seven hundred and thirty-six human cases of the disease have been confirmed only in São Paulo state from 2011 to 2020, with a mortality rate of 63.5% [3]. It is, therefore, imperative to control the capybara population by removing individuals [4] or using contraceptive measures [5]; these are strategies that may require capture procedures [5–8].

**Funding:** Jefferson Farias Cordeiro had received financial support from the Fundação de Amparo à Pesquisa do Estado de São Paulo - FAPESP, (2016/ 21121-9).

**Competing interests:** NO authors have competing interests.

Several anaesthetic protocols for captive or free-ranging capybaras have been previously described [6–12], but no study has assessed the impact of these protocols on the arterial partial pressure of oxygen (PaO$_2$). Hypoxaemia is a concern during the chemical restraint of wild animals [13] as it can lead to myopathy and myocardial hypoxia [14, 15]. Thus, oxygen (O$_2$) supplementation has been used to prevent or treat hypoxaemia during the immobilisation of wild animals [13, 16–19]. Furthermore, chemical restraint protocols in capybaras have been based on dissociative anaesthesia (ketamine or tiletamine) in combination with other drugs (xylazine, romifidine, midazolam, zolazepam, levomepromazine, medetomidine, or dexmedetomidine) [7–12], and may be associated with muscle spasticity, nystagmus, disorientation, increased heart rate, and blood pressure [9]. In general, if the state of 'dissociation' persists during recovery, it can cause disorientation, anxiety, stress, myoclonus, catalepsy, and abnormal behaviour in the targeted animal [20].

Reversible chemical restraint combination protocols have been used in free-ranging animals, seeking to avoid the side effects observed with dissociative anaesthesia [18, 19, 21, 22]; the advantages of their use include animals returning to their normal pattern of activities quicker and safer, preventing them from becoming vulnerable to predators or accidents due to the disorientation related to anaesthetic recovery. Similarly, they can given drug reversal agents if a dose higher than the recommended dose is administered or if the animal experiences unwanted side effects.

The objectives of this study were to assess the impact of fully reversible chemical restraint protocols on free-ranging capybaras and the effects of different O$_2$ supplementation flows. Based on the studies carried out with other species [18, 19, 23] as well as the authors' experience, we hypothesised that nasal oxygen flows of either 2 or 5 L/min would reverse hypoxaemia in chemically restrained capybaras.

## Materials and methods

The study was approved by the Chico Mendes Institute for Biodiversity Conservation (protocol 58028) and the Animal Ethics Committee of the Faculty of Animal Science and Food Engineering, University of São Paulo (protocol 9796180717). The animals were captured between May 2018 and October 2019 in southeast Brazil (21˚59´46˝S; 47˚25´33˝W) at 627 m above sea level during rainless nights with an average temperature of 21.8 ± 3.5˚C.

To capture the capybaras, two portable corral-style traps (4 and 25 m$^2$, respectively) with an iron fence or metal mesh walls were used and baited with corn or sugarcane in automatic closing platforms. If more than one animal was caught at once, only one animal was immobilised, and the others were released. Capybaras were excluded if they were very wet, weighed less than 20 kg, had severe injuries, or did not allow manipulation after 20 min after the drug injection. Thirty-six of 49 capybaras that were captured were included in the study; these included 30 females and 6 males weighing 48.9 ± 17.8 kg (actual body weight).

After the capybaras were caught in the corral-style trap, they were physically restrained with a one-metre diameter net for drug administration. They were given one of two drug combinations by intramuscular hand injection in the lateral side of the hind limb using a syringe and a 20 G needle. The drug combinations used were 5 μg/kg dexmedetomidine (Dexdomitor® 0.5 mg/mL, Zoetis, São Paulo, SP, Brazil) + 0.1 mg/kg midazolam (Dormire® 5 mg/mL, Cristalia, Itapira, SP, Brazil) + 0.2 mg/kg butorphanol (Torbugesic® 10 mg/mL, Fort Dodge Animal Health, Fort Dodge, IA, USA) (DMB, n = 18) or 5 μg/kg dexmedetomidine + 0.1 mg/kg midazolam + 0.1 mg/kg methadone (Mytedom® 10 mg/mL, Cristalia, Itapira, SP, Brazil) (DMM, n = 18). The doses were calculated based on the estimated body weights. The animals were weighed during immobilisation, after which they were kept in right lateral recumbency.

Six animals from each drug protocol group were maintained in breathing air (21% $O_2$) (DMB-Air and DMM-Air) throughout the procedure, whereas six animals received $O_2$ supplementation at 2 L/min (DMB-2L and DMM-2L) and six others received $O_2$ supplementation at 5 L/min (DMB-5L and DMM-5L) for 30 min after the initial drug injection. This design was proposed to understand which treatment might reverse a possible hypoxaemia condition, when $PaO_2 < 80$ mmHg. Oxygen was provided from a 5-L portable cylinder connected to a number 16 bladder probe (outer diameter 5.3 mm) lubricated with 2% lidocaine (Xylestesin jelly 20 mg/mL, Cristalia, Itapira, SP, Brazil) and introduced into one of the nostrils up to the medial canthus of the eye. Oxygen supplementation was provided for 30 min and discontinued 60 min after the drug injection. Afterward, the immobilisation was reversed with an intramuscular injection of 2 μg/kg flumazenil (Flumazenil 0.1 mg/mL, Cristalia, Itapira, SP, Brazil) + 5 μg/kg atipamezole (Antisedan 5 mg/mL, Zoetis, São Paulo, SP, Brazil) + 4 μg/kg naloxone (Narcan 0.4 mg/mL, Cristalia, Itapira, SP, Brazil), mixed in the same syringe, and the animals were released from the traps 30 min later. The treatment order was assigned as a block randomisation design, using a website (www.sorteador.com.br).

Respiratory rate (RR) was monitored by observing chest movements, and heart rate (HR) was measured by auscultation of the heart. Arterial blood samples were withdrawn from a branch of the femoral artery with a heparinised 1-mL syringe and a 22 G needle for blood gas analysis and the analysis of other blood variables. The first sample was withdrawn 30 min after the injection, when all the animals were breathing air, and the second sample was withdrawn 60 min after the injection. The samples were immediately processed with a portable blood gas analyser (I-Stat®1 and CG8+ Cartridges, Abbott Point of Care, IL, USA). pH, $PaO_2$, partial pressure of arterial $CO_2$ ($PaCO_2$), plasma ionised calcium ($Ca^{2+}$), potassium ($K^+$), sodium ($Na^+$), and glucose concentrations were measured, and the bicarbonate ($HCO_3^-$) and haemoglobin oxygen saturation ($SaO_2$) were calculated. The results were corrected for the rectal temperature of the animal. The ratio of arterial oxygen partial pressure to fractional inspired oxygen ($PaO_2/F_iO_2$) was also calculated.

Plasma lactate was measured with another handheld device (Accutrend® Plus—Roche Diagnostics, Mannheim, Germany). The device had a lower limit of detection of 0.8 mmol/L, and values below that were considered rounded to 0.7 mmol/L for statistical comparison.

The alveolar-arterial $O_2$ gradient [$P(A-a)O_2$], $PaO_2$, and partial pressure of inspired $O_2$ ($PiO_2$) were estimated at a standard temperature of 37˚C, respiratory coefficient (RQ) of 1, barometric pressure ($P_B$) of 707.5 mmHg, fraction of inspired $O_2$ ($F_iO_2$) of 0.21, and saturated water vapour pressure ($P_{H2O}$) of 47 mmHg.

$$[P(A-a)O_2] = PAO_2 - PaO_2$$
$$PAO_2 = P_iO_2 - (PaCO_2/RQ)$$
$$P_iO_2 = F_iO_2 \times (P_B - P_{H_2O})$$

Based on the principle that $[P(A-a)O_2] < 10$ mmHg, the minimum expected normal value for $PaO_2$ was calculated for animals breathing ambient air.

$$PaO_{2 \text{ minimum expected}} > PAO_2 - 10$$

The statistical analysis was performed using GraphPad Prism Software (San Diego, CA, USA). The variables (mean ± standard deviation) were considered parametric if they were normally distributed according to the Shapiro–Wilk test and had a coefficient of variation of less than 0.2; otherwise, they were considered nonparametric data (median [interquartile range]). For parametric data, a one-tailed unpaired t-test was used for comparison of values at 30 and 60 min, and analysis of variance (ANOVA) with Tukey's post-test was performed for

comparing data among the treatments. For non-parametric data, Wilcoxon test and Kruskal–Wallis test with Dunn's post-test were used. Pearson's correlation analysis was used to assess the correlation between the impact of O$_2$ supplementation and blood gas variables at 60 min while comparing the animals of the Air group with those of the 2 L/min and 5 L/min groups. Differences were considered statistically significant at $p < 0.05$.

## Results

The capybaras remained quiet after they were caught in the trap. Both DMM (9.2 ± 3.0 min) and DMB (10.5 ± 3.7 min) produced an equally fast induction, and there were no deaths during the study. No statistically significant differences were observed between the DMB and DMM groups. Thus, the animals from the DMB and DMM were grouped together for a more robust data analysis.

HR was approximately 70–80 beats/min and RR was about 26–36 breaths/min throughout the procedure. There was no statistical difference in these variables among the treatments. The same was observed in the arterial blood pH, over time. All animals that were breathing air only developed hypoxaemia at 30 min after drug injection, and remained hypoxaemic at 60 min. PaO$_2$ was lower than the PaO$_{2\text{minimum expected}}$, i.e., higher than 83 ± 4 mmHg while breathing air, resulting in a P(A-a)O$_2$ of 49 [16; 65] mmHg and PaO$_2$/FiO$_2$ ratio of 209 [190; 228]. After O$_2$ supplementation at a flow rate of 2 L/min, there was an increase in SaO$_2$ and PaO$_2$ compared with that at 30 min when breathing only air ($p < 0.0001$; $p = 0.0002$, respectively), but 9 of the 12 capybaras remained hypoxaemic, and there was no significant difference in both variables compared with the air group at 60 min (Table 1). A flow rate of 5 L/min reversed the hypoxaemia in all 12 animals (PaO$_2$, $p < 0.0001$) (Table 1).

The PaCO$_2$, HCO$_3^-$, and plasma glucose levels increased significantly over time (from 30 min pre-O$_2$ to 60 min) in the 5 L/min group ($p = <0,0001$) (Table 1). There was difference in plasma lactate over time only in the 5 L/min group ($p = 0.0365$). Values above 2 mmol/L were measured in six animals in the Air group, four animals in the 2 L/min group, and none in the 5 L/min group. Rectal temperature decreased over time in the 5 L/min group ($p < 0.0001$) (Table 1). Regarding electrolytes, ionised Ca$^{2+}$ concentration slightly decreased in the Air group between 30 and 60 min ($p = 0.0449$) and increased in the 5L/min group ($p = 0.030$), K$^+$ concentration increased in the 2L/min group ($p = 0.0161$), while Na$^+$ did not vary over time in any group (Table 1).

Comparing the data from the capybaras belonging to the Air and 2 L/min groups at 60 min, positive correlations were found between increased flow and SaO$_2$ and PaO$_2$. Comparing the air and 5 L/min groups at 60 min, positive correlations were observed between the flow and SaO$_2$, PaO$_2$, PaCO$_2$, HCO$_3^-$, Ca$^{2+}$, and glucose. The increase in the O$_2$ flow rate was negatively correlated with lactate (Fig 1).

## Discussion

The chemical restraint of the free-ranging capybaras is an important step in population management, and optimal maintenance of respiratory capacity is indispensable in choosing the anaesthetic protocol. Thus, we assessed the changes in the respiratory variables and electrolytes in the capybaras after the administration of two different drug combinations, based on a benzodiazepine, an alpha$_2$ adrenergic agonist, and an opioid. We also assessed the effectiveness of two O$_2$ supplementation flows and the factors involved in this process.

Although blood gas analysers use algorithms based on human haemoglobin, these values have commonly been used in species for which results validation studies have not been carried

**Table 1. Heart rate, respiratory rate, blood gas analysis, rectal temperature, glucose, lactate, plasma ionised calcium (Ca$^{2+}$), sodium (Na$^+$) and potassium (K$^+$) plasma concentrations in free-ranging capybaras (*Hydrochoerus hydrochaeris*) chemically restrained with dexmedetomidine and midazolam, and either butorphanol (DMB) or methadone (DMM)§.** All animals (n = 36) were breathing air when the 30-min sample was collected (pre-treatment) and supplemented nasally or not with oxygen after that until the 60-min sample was collected. Values expressed as mean ± standard deviation or median [interquartile range].

| Variable | Pre-treatment | 30 min | Treatment | 60 min |
|---|---|---|---|---|
| HR | Air (n = 36) | 79 ± 8.8 | Air (n = 12) | 76 ± 5.8 |
| | | | 2L (n = 12) | 74 ± 10.9 |
| | | | 5L (n = 12) | 73 ± 4.5 |
| RR | Air (n = 36) | 36 [20; 55] | Air (n = 12) | 30 [25; 38] |
| | | | 2L (n = 12) | 34 [25; 47] |
| | | | 5L (n = 12) | 26 [21; 32] |
| pH | Air (n = 36) | 7.40 ± 0.04 | Air (n = 12) | 7.42 ± 0.04 |
| | | | 2L (n = 12) | 7.40 ± 0.04 |
| | | | 5L (n = 12) | 7.40 ± 0.02 |
| PaO$_2$ (mmHg) | Air (n = 36) | 44 [40; 47] | Air (n = 12) | 51 [45; 56][A] |
| | | | 2L (n = 12) | 63 [56; 86][*A] |
| | | | 5L (n = 12) | 188 [171; 286][*B] |
| SaO$_2$ (%) | Air (n = 36) | 81.6 ± 6.1 | Air (n = 12) | 85.7 ± 6.0[A] |
| | | | 2L (n = 12) | 92.4 ± 4.2[*A] |
| | | | 5L (n = 12) | 99.8 ± 0.4[*B] |
| PaCO$_2$ (mmHg) | Air (n = 36) | 45 ± 4 | Air (n = 12) | 45 ± 4[A] |
| | | | 2L (n = 12) | 46 ± 7[A] |
| | | | 5L (n = 12) | 54 ± 4[*B] |
| HCO$_3^-$ (mEq/L) | Air (n = 36) | 29 [28; 31] | Air (n = 12) | 30 [26; 32][A] |
| | | | 2L (n = 12) | 29 [26; 31][A] |
| | | | 5L (n = 12) | 35 [34; 36][*B] |
| Rectal temp. | Air (n = 36) | 36.5 [35.7; 36.8] | Air (n = 12) | 36.6 [36.2; 37.2][A] |
| | | | 2L (n = 12) | 36.2 [35.6; 36.8][A] |
| | | | 5L (n = 12) | 35.2 [35.0; 35.7][*B] |
| Glucose (mg/dL) | Air (n = 36) | 197 [169; 214] | Air (n = 12) | 206 [162; 224][A] |
| | | | 2L (n = 12) | 188 [180; 191][A] |
| | | | 5L (n = 12) | 273 [245; 301][*B] |
| Lactate (mmol/L) | Air (n = 36) | 0.75 [0.70; 2.07] | Air (n = 12) | 2.10 [0.70; 3.65] |
| | | | 2L (n = 12) | 0.85 [0.70; 2.58] |
| | | | 5L (n = 12) | 0.70 [0.70; 0.77][*] |
| Ca$^{2+}$ (mEq/L) | Air (n = 36) | 0.95 [0.86; 1.09] | Air (n = 12) | 0.85 [0.65; 0.96][*A] |
| | | | 2L (n = 12) | 0.91 [0.70; 1.06][AB] |
| | | | 5L (n = 12) | 1.15 [0.95; 1.19][*B] |
| Na$^+$ (mEq/L) | Air (n = 36) | 135.5 ± 1.96 | Air (n = 12) | 135.92 ± 1.38 |
| | | | 2L (n = 12) | 135.22 ± 3.16 |
| | | | 5L (n = 12) | 136.17 ± 1.75 |
| K$^+$ (mEq/L) | Air (n = 36) | 3.9 ± 0.43 | Air (n = 12) | 4.18 ± 0.60 |
| | | | 2L (n = 12) | 4.33 ± 0.61[*] |
| | | | 5L (n = 12) | 3.93 ± 0.36 |

HR, heart rate; RR, respiratory rate; pH, potential hydrogen; PaO$_2$, partial pressure of arterial of oxygen; SaO$_2$, haemoglobin oxygen saturation; PaCO$_2$, partial pressure of arterial of carbon dioxide; HCO$_3^-$, bicarbonate; Ca$^{2+}$, plasma ionised calcium; K$^+$, plasma potassium; Na$^+$, plasma sodium; mmHg, millimetres of mercury; mEq/L, milliequivalents per litre; mmol/L, millimoles per litre; mg/dL, milligrams per decilitre; Air, animals without oxygen supplementation; 2L, animals which received 2L/min oxygen supplementation after 30 min; 5L, animals which received 5L/min oxygen supplementation after 30 min.

* indicates a significant difference between the times (30 and 60 min), and different superscript letters indicate differences between the treatments ($p < 0.05$). pH, PaO$_2$ and PaCO$_2$ values were corrected to the rectal temperature of the animal.

§ The animals from the DMB and DMM were grouped together for a more robust data analysis since there were no statistical differences between the groups.

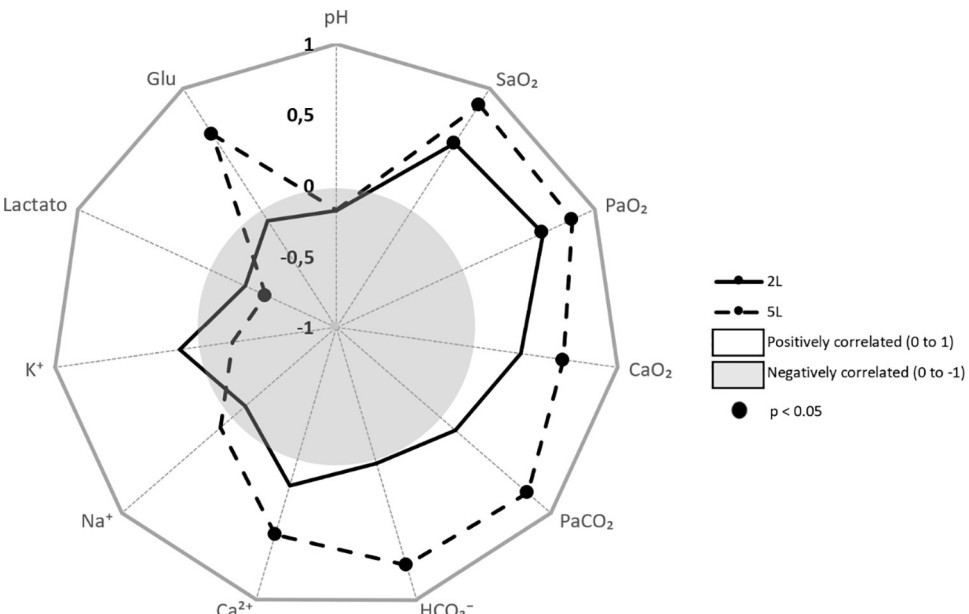

**Fig 1. Pearson's correlation coefficient (r) and significance (p) of blood variables with different O₂ flows during immobilisation of free-ranging capybaras (*Hydrochoerus hydrochaeris*).** pH, potential hydrogen; SaO₂, haemoglobin oxygen saturation; PaO₂, partial pressure of arterial of oxygen; PaCO₂, partial pressure of arterial of carbon dioxide; HCO₃⁻, bicarbonate; Ca²⁺, plasma ionised calcium; Na⁺, plasma sodium; K⁺, plasma potassium; Glu, glucose; 2L, animals which received 2L/min oxygen supplementation after 30 min until 60 min; 5L, animals which received 5L/min oxygen supplementation after 30 min until 60 min. Vertical axis represents *r* values. In the bright area, the closer to the edge, the greater the positive correlation. In the dark area, the closer to the centre, the greater the negative correlation. Markers represent a significant correlation (p <0.05).

out [17–19, 24]. Thus, the data obtained here for capybaras can help in interpreting the results and variations observed due to the change in oxygen flow.

Arterial blood gases did not differ when methadone or butorphanol was included in the drug protocol. Due to the agonist function of methadone at the μ-opioid receptors, it was expected that methadone would induce greater respiratory depression than butorphanol, since the latter acts as an antagonist at the μ-opioid receptors [25]. However, our results are consistent with those of another study that reported no difference in sheep sedated with a combination of dexmedetomidine and butorphanol or methadone [26].

Blood gas analysis is crucial for detecting hypoxaemia in capybaras. All animals developed hypoxaemia at 30 min, even in the absence of evident clinical signs. Although the portable blood gas analyser used was based on human-determined algorithms, it was and is, together with the co-oximetre, one of the available tools for measuring oxygen levels for field conditions [27]. In addition to the parameters of an ordinary blood gas analysis, there are also easy-to-calculate indices that allow a better understanding of respiratory dynamics and provide clinically useful prognostic information. It was found that the values of $[P(A\text{-}a)O_2]$ were higher than 10 mmHg and the $PaO_2/F_iO_2$ ratio was less than 300 in animals breathing air. This was also observed during the chemical restraint of other wild animals, indicating that changes may have occurred in the integrity of the alveolar capillary, causing hypoxaemia due to intrapulmonary factors, such as the formation of shunts and ventilation/perfusion imbalance [17, 28, 29].

Although the intranasal flow of 2 L/min O₂ improved the PaO₂ in all animals, nine animals remained hypoxaemic. In comparison, the low flow rate of 2 L/min O₂ was adequate to

prevent hypoxaemia in cheetahs (*Acinonyx jubatus*) sedated with dexmedetomidine, butorphanol, and midazolam [18] and revert hypoxaemia in brown bears (*Ursus arctos*) immobilised with the medetomidine/tiletamine-zolazepam combination [23].

Hypercapnia (PaCO$_2$ > 45 mmHg) was observed in six animals that were breathing air, six in the 2 L/min group, and all the 12 in the 5 L/min group at 60 min. Hypercapnia may have occurred due to poor ventilation, but its increase in the 5 L/min group may be due to the inhibition of hypoxic pulmonary vasoconstriction. This compensatory mechanism is activated in response to low P$_A$O$_2$ and diverts blood flow to better-ventilated areas of the lung [30–32]. Thus, the CO$_2$ produced goes through the functional alveoli before it is eliminated. When hypoxaemia was reversed with O$_2$ supplementation, hypoxic pulmonary vasoconstriction was inhibited, leading to impaired CO$_2$ elimination.

The Haldane effect can also be involved in this process, as higher levels of O$_2$ acidify haemoglobin, which impairs the binding of CO$_2$ and increases PaCO$_2$ [33]. Although the F$_i$O$_2$ of animals was not measured in this study, it is known that high F$_i$O$_2$ rates are associated with the appearance of atelectasis, and if they are provided to individuals with low ventilation/perfusion ratio, the lungs may collapse because alveolar gas passes into the blood at a higher flow rate than that at which it is inspired [34].

High values of plasma glucose were expected because glycogenolysis occurs to provide energy to meet the increased energy demand during the increase in cardiac frequency during stressful conditions, such as capture [35]. In addition, dexmedetomidine decreases the levels of insulin [36], which may lead to an increase in plasma glucose within a few minutes after drug administration [37]. However, plasma glucose increased over time only in the 5 L/min group. Anaerobic glycolysis was probably present in some animals in the air and 2 L/min groups due to low O$_2$ levels. Despite the highest level of lactate (9.9 mmol/L) recorded in a capybara that was breathing air at 60 min, lactic acidosis was not observed.

## Conclusion

Both drug protocols quickly induced chemical immobilisation in the capybaras, but they also caused hypoxaemia, regardless of the opioid used. Furthermore, oxygen supplementation should be provided to prevent hypoxaemia during chemical restraint, when these drug combinations are used. Based on the results of this study, a flow rate of 5 L/min is recommended because the lower evaluated flow rate of 2 L/min did not correct the hypoxaemia in all the animals. However, further studies are needed to determine if 3 or 4 L/min is adequate.

## Acknowledgments

We would like to thank Thais Ferres Bressan, Milena Fascina Bovi, Roberto Romano do Prado Filho and Wellington Henrique Bessi for technical support.

## Author Contributions

**Conceptualization:** Jefferson F. Cordeiro, Adriano B. Carregaro.

**Data curation:** Jefferson F. Cordeiro, Mariana C. Sanches, Elidiane Rusch, Nathalia V. Xavier, Ana Angélica Cassoli, Adriano B. Carregaro.

**Formal analysis:** Jefferson F. Cordeiro, Åsa Fahlman, Adriano B. Carregaro.

**Funding acquisition:** Adriano B. Carregaro.

**Investigation:** Jefferson F. Cordeiro, Mariana C. Sanches, Elidiane Rusch, Nathalia V. Xavier, Ana Angélica Cassoli, Adriano B. Carregaro.

**Methodology:** Jefferson F. Cordeiro, Mariana C. Sanches, Elidiane Rusch, Nathalia V. Xavier, Ana Angélica Cassoli, Adriano B. Carregaro.

**Project administration:** Adriano B. Carregaro.

**Supervision:** Adriano B. Carregaro.

**Writing – original draft:** Jefferson F. Cordeiro, Mariana C. Sanches, Elidiane Rusch, Nathalia V. Xavier, Ana Angélica Cassoli.

**Writing – review & editing:** Jefferson F. Cordeiro, Åsa Fahlman, Adriano B. Carregaro.

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
