## [Decision Letter · Decision Letter 0]

9 Jul 2021

PONE-D-21-16924

Intranasal oxygen reverses hypoxaemia in immobilised free-ranging capybaras (Hydrochoerus hydrochaeris)

PLOS ONE

Dear Dr. Carregaro,

Thank you for submitting your manuscript to PLOS ONE. After careful consideration, we feel that it has merit but does not fully meet PLOS ONE’s publication criteria as it currently stands. Therefore, we invite you to submit a revised version of the manuscript that addresses the points raised during the review process.

As you can notice from the reviewers comments below, your manuscript received two discrepant reviews in terms of the suitability for PLOS ONE. My own expertise in this area are limited and, as such, my decision regarding the major revisions is simply compromise between the two reviews. In revising your manuscript I recommend you pay considerable attention to the general as well as specific comments provided by reviewer two. 

We look forward to receiving your revised manuscript.

Kind regards,

Ruud AW Veldhuizen

Academic Editor

PLOS ONE

2. Please expand the acronym “J.F.C” and “FAPESP” (as indicated in your financial disclosure) so that it states the name of your funders in full.

3. Thank you for stating the following in the Acknowledgments/Funding Section of your manuscript:

“This research was funded by Fundação de Amparo à Pesquisa do Estado de São Paulo - FAPESP, No. 2016/21121-9.”

Funding information should not appear in the Acknowledgments section or other areas of your manuscript. We will only publish funding information present in the Funding Statement section of the online submission form.

“This research was funded by Fundação de Amparo à Pesquisa do Estado de São Paulo - FAPESP, No. 2016/21121-9.”

Additional Editor Comments (if provided):

Reviewers' comments:

Reviewer's Responses to Questions

**Comments to the Author**

1. Is the manuscript technically sound, and do the data support the conclusions?

Reviewer #1: Yes

Reviewer #2: Yes

2. Has the statistical analysis been performed appropriately and rigorously? 

Reviewer #1: I Don't Know

Reviewer #2: Yes

3. Have the authors made all data underlying the findings in their manuscript fully available?

Reviewer #1: Yes

Reviewer #2: Yes

4. Is the manuscript presented in an intelligible fashion and written in standard English?

Reviewer #1: Yes

Reviewer #2: Yes

5. Review Comments to the Author

Reviewer #1: 1. Line 26 : Mention the scientific name

2. Line 90 : Describe the recumbency in which the animals were maintained

3. Line 90 : ...... maintained in breathing air.... (Mention the percentage of oxygen for better understanding)

4. Line 93: Oxygen was provided ............ (Mention the percentage of oxygen for better understanding)

5. Line 96:..... height of medial corner of the eye (Consider using the term medial canthus of eye)

6. Line 98 to 102: Was the reversal agents mixed in a single syringe ?? (Describe it as in separate syringes)

Reviewer #2: Dear authors,

this study aims to verification of hypoxaemia prevention in capybaras by oxygen supplementation during anesthesia. As any anesthetic protocol for wild animals the presented findings may be of use in wildlife population control as well as for treatment of animals in captivity. This makes it a worth publishing. On the other hand, the paper is kept very technical. Without any additional contribution to the field and I do not find it suitable for publication in high ranking journal in its current state. I am missing a review of so far published anesthetic protocols in capybaras and I do not like the experimental design. I would understand if the design was governed by other study/procedure and this study was a by product, but there is no indication for that. Moreover the most basic physiological parameters - breathing rate and hearth rate - were not measured. Apart from this, I have also following specific comments:

45 "it can" -> "it's population"

54 there are a multiple anesthesia protocols missing. Eg https://doi.org/10.29374/2527-2179.bjvm107220, https://www.ncbi.nlm.nih.gov/pmc/articles/PMC7136804/ or https://www.jstor.org/stable/3784559.

55-58 Please show why is this a concern with capybaras. Do other protocols result in hypoxaemia as well? Even anecdotic observations could be informative. It would be good if you could justify why you used a different anesthetic protocol than those already published. Are there any practical or theoretical advantages or disadvantages of your protocol or of the published protocols?

59-62 Please explain the choice of oxygen flow rates. Did you have some preliminary data?

Was any other treatment done on the capybaras (eg sterilization) during the anesthesia or was the restraint only because of this study?

Materials and methods

78 please indicate the sex in the raw data and check the (rounding of) mean weight. My calculation suggest that the average mass is 49kg.

82-88 Why did you choose these exact doses? Are these doses recommended by the manufacturer for rodents/caviomorphs/capybaras?

88 Was weight estimated or measured? If estimated, please add the dose of anesthetic in the raw data.

90-93 Please justify the design. I do not see apparent advantage in your design. It seems to me that it would be better if you started started with 3 groups, which would either be air, 2L/min and 5L/min oxygen supplementation immediately after onset on anesthesia. Such design would allow more direct comparison by paired tests and would also be more linked with the planned use of the anesthesia protocol. I do not expect anyone to first wait 30 min and then apply oxygen to follow your protocol. If you aimed to show that hypoxaemia is reversible, please state this clearly.

103 "all the groups ..." please be more specific here. I do not understand the sentence, what repetition you mean?

106 "25x7 mm" Please specify this in more detail or remove the information.

111 Is there any support for the assumption that the cartridges measure/calculate SaO2 and other parameter correctly? I doubt that there is data for this in capybaras and as such this should be clearly admitted. I do not find your statement in discussion too helpful. You could for example estimate the possible error or put emphasis on the non-calculated parameters. What is the precision of measurement by CG8+ cartridges?

113 How was the Tb measured?

116 I believe that this is not correct. Do you have any justification for this approach?

120 Why didn't you use the measured body temperature?

120 RQ=1 - Why? Normally assumed RQ is 0.85.

122 pH2O=47 mmHg - please justify this assumption. It seems to me that you assume that the temperature of the expired air has temperature of 37°C and saturated water vapour content at that temperature. I doubt this is correct. The maxilloturbinals in the nasal cavity will for sure preserve some heat and water, resulting in lower expired air temperature and lower water vapor pressure in the expired air. Also please explain all abbreviations. PAO2 and PiO2 are not explained.

136 Why did you use one-tailed paired t-test? For single tailed test you need to state the hypothesis first (greater/lower). What do you mean by intragroup?

Results

146 "The capybaras remained quiet after they were caught in the trap." How do you know this? Were the traps observed?

146 "The small size of the traps facilitated physical restraint with the net for drug administration by hand injection." This belongs to methods.

148 not meaningful to pool the two protocols at this moment. You explain the pooling below. Please check the mean, my calculations show 9.9min mean induction time.

149-150. missing Test statistics, p levels, df. Pleas provide this data and the tests for the revision process or even better place them in an appendix.

153 Please define hypoxaemia in methods. What is the threshold for hypoxaemia?

154 This information is not shown in Table 1. Only means are presented in Table 1. Please reformulate the sentence.

154 check/remove the ">" sign.

157 difference in what?

160 Please list all abbreviation in the table caption. Table should be able to stand alone. Valid for Tab. 2 as well.

Tab 1 3rd column. Group does not seem to be a good name. Moreover this column is the same in whole table and therefore redundant. Please remove. Same for table 2

Please list the parameters by subgroups as well. Add subgroup term in methods, you call call them groups there You may still keep the grand mean as well. You are showing statistical significant differences for the paired test and we do not see the first value of the pair. Moreover, it seems that you are using paired t-test, which has an assumption of normal distribution for data which you tested distribution and claim non-normal for a different test.

SaO2 It seems that you compared percentages by parametric test without any transformation. Please justify, the usual apprach is to perform arcsine transformation prior to statistical comparison.

PaO2/FiO2 please add to methods how was this calculated. Add a unit. And consider removing from the table as it is not used in the second part of the table.

P(A-a)O2 Consider removing from the table or filling the other part of the table.

175 this wording suggest gradual increase, but only two measurements do not allow this statement.

Table 2

Why is this table separate from table 1? Consider merging the tables.

Unify the units throughout the text. Please express the units per Liter or justify. I also suggest to use SI units, ie mol/L everywhere where possible.

Lactate: please check if median really equals lower boundary of interquartile range. The raw data suggest median of 1 and not 0.7 in the 30 min group. Please check the other values as well. Moreover the numbers do not seem to be correct because of the assumption on line 116.

Na+ and K+ data missing for one animal in the raw data and the methods state that animals with missing data were excluded. Please adjust.

200 Fig 1. Consider erasing the figure and putting the correlations in the table1/2. "BE" not explained anywhere in the text, please delete it from the figure. In the image legend on the right change the decimal comma to decimal point. Remove the legend bellow the figure and put the text in figure caption.

204 Was the p-value adjusted for multiple tests? if yes, please add this information to methods. If not, consider doing so. With 22 test the probability of false positive is very high.

222 patients?

226 But how reliable and comparable with reality these figures are?

228-230. Ratios do not have dimension. Additionally the sentence does not have verb and is meaningless, please reformulate.

234 It seems to me that PaO2 increased in ALL animals in 2L group. Moreover I suggest not to mix PaO2 and hypoxaemic. Please be careful in formulations.

239 "animals that were breathing air in the 2L group" is not clear. Two meanings are possible. Either delete the breathing air or 2L group.

259 why however?

260 please check the units. You are using mmol/L here, but in table 2 you are using mg/dL with the same parameter. One of these is probably faulty.

265 If no long-term adverse effect of the anesthesia on the animals was observed, how necessary is to prevent hypoxaemia then?

6. PLOS authors have the option to publish the peer review history of their article (what does this mean?). If published, this will include your full peer review and any attached files.

Reviewer #1: No

Reviewer #2: No

---

## [Author Response · Author response to Decision Letter 0]

22 Sep 2021

REVIEWER 1

- Line 26: Mention the scientific name.

- Line 90: Describe the recumbency in which the animals were maintained.

- Line 90: ...... maintained in breathing air.... (Mention the percentage of oxygen for better understanding).

- Line 93: Oxygen was provided ............ (Mention the percentage of oxygen for better understanding).

Answer: Thank you for this remark. Information has been added (lines 26, 103-111).

- Line 96: “height of medial corner of the eye (Consider using the term medial canthus of eye)

Answer: Thank you for this remark. Information has been included (lines 114-115).

- Line 98 to 102: Was the reversal agents mixed in a single syringe?

Answer: Thank you for the comment. Capybaras were given a mixture of the reversal agents. This information was added in this revised version in lines 119-120.

REVIEWER 2

- This study aims to verification of hypoxaemia prevention in capybaras by oxygen supplementation during anesthesia. As any anesthetic protocol for wild animals the presented findings may be of use in wildlife population control as well as for treatment of animals in captivity. This makes it a worth publishing. On the other hand, the paper is kept very technical. Without any additional contribution to the field and I do not find it suitable for publication in high ranking journal in its current state. I am missing a review of so far published anesthetic protocols in capybaras and I do not like the experimental design. I would understand if the design was governed by other study/procedure and this study was a by product, but there is no indication for that. Moreover the most basic physiological parameters - breathing rate and hearth rate - were not measured. 

Answer: We thank the Reviewer for the comments and to fit this paper as a technical study. It was one of our propose. The authors have been studying about chemical restraint techniques in exotics and their impact on this procedure in different species. It is well-known hypoxaemia is the main issue in exotics capture, which is poorly identified and treated as well. Thus, the main focus of this study was to deliver a total reversible chemical restraint protocol, safer and avoiding hypoxaemia during the procedure. Indeed, the HR and the RR were measured throughout the procedure. They were not described in the original version because we have been writing down another paper, comparing cardiorespiratory monitoring, induction time, immobilization and recovery observed with both protocols. However, we agree with de Reviewer about the importance to describe HR and RR. Thus, as requested, we decided to add them as one mean/median in this revised version (lines 122-123, 173-174). The Reviewer comments certainly enriched our study.

- line 45 "it can" -> "it's population"

Answer: Thank you for this remark. The text has been changed in this revised version (lines 46-47).

- line 54 there are a multiple anesthesia protocols missing. Eg https://doi.org/10.29374/2527-2179.bjvm107220, https://www.ncbi.nlm.nih.gov/pmc/articles/PMC7136804/ or https://www.jstor.org/stable/3784559.

Answer: Thank you for the comment. We would like to point out the first paper was published 13 days before we had submitted our MS. So, it was not possible to add it to that original version. The second one is a review, not a study, and the last one focused on methods of capture, with poor description about drugs, monitoring and other physiological information. Thus, we have decided to add only the Rosenfield et al study in this revised version (ref #12, line 55).

- lines 55-58 Please show why is this a concern with capybaras. Do other protocols result in hypoxaemia as well? Even anecdotic observations could be informative. It would be good if you could justify why you used a different anesthetic protocol than those already published. Are there any practical or theoretical advantages or disadvantages of your protocol or of the published protocols?

Answer: Thank you for this remark. It is well-known hypoxaemia is the main issue in exotics chemical restraint procedures. As mentioned in the paper, no studies aimed at blood gas analysis in capybaras, neither one that assesses hypoxaemia and oxygen supplementation. The disadvantages of ordinary protocols, based on dissociative anaesthesia, and the advantages of the reversible chemical restraint protocols have now been highlighted in the introduction (lines 59-73).

- lines 59-62 Please explain the choice of oxygen flow rates. Did you have some preliminary data?

Answer: Thank you for this remark. The information was added in the revised version (lines 76-77). 

- Was any other treatment done on the capybaras (eg sterilization) during the anesthesia or was the restraint only because of this study?

Answer: No. They were caught only for this study.

- line 78: please indicate the sex in the raw data and check the (rounding of) mean weight. My calculation suggest that the average mass is 49kg.

- line 88: Was weight estimated or measured? If estimated, please add the dose of anesthetic in the raw data.

Answer: Thank you for the remark. Sex was added in the raw data. We apologize about the mean weight. We added one digit after decimal number. This information was corrected in this revised version (line 94).

- lines 82-88: Why did you choose these exact doses? Are these doses recommended by the manufacturer for rodents/caviomorphs/capybaras?

Answer: Thank you for the comment. The doses were adjusted from pilot studies.

- lines 90-93: Please justify the design. I do not see apparent advantage in your design. It seems to me that it would be better if you started with 3 groups, which would either be air, 2L/min and 5L/min oxygen supplementation immediately after onset on anesthesia. Such design would allow more direct comparison by paired tests and would also be more linked with the planned use of the anesthesia protocol. I do not expect anyone to first wait 30 min and then apply oxygen to follow your protocol. If you aimed to show that hypoxaemia is reversible, please state this clearly.

Answer: Thank you for this remark. This suggestion could be done, indeed, but we were seeking the impact of the O2 supplementation to reverse the hypoxemia. Thus, we must let them breathing air, to evaluate if they developed a low PaO2, and then, give them the treatment. The text was rewritten in this revised version (lines 106-111). 

- line 103: "all the groups ..." please be more specific here. I do not understand the sentence, what repetition you mean?

Answer: The text was rewritten in this revised version (lines 120-121).

- line 106: "25x7 mm" Please specify this in more detail or remove the information.

Answer: Thank you for this remark. The text was rewritten in this revised version (line 125).

- line 111: Is there any support for the assumption that the cartridges measure/calculate SaO2 and other parameter correctly? I doubt that there is data for this in capybaras and as such this should be clearly admitted. I do not find your statement in discussion too helpful. You could for example estimate the possible error or put emphasis on the non-calculated parameters. What is the precision of measurement by CG8+ cartridges?

Answer: Although blood gas analysers use algorithms based on human haemoglobin, these values have commonly been used in species for which results validation studies have not been carried out.

- line 113 How was the Tb measured?

Answer: We sorry about that but what does Tb mean?

- line 116: I believe that this is not correct. Do you have any justification for this approach?

Answer: The measurement of serum lactate under field conditions was one of the limitations of the study. According to the manufacturer's specifications, the lower limit of the measuring range of the device is 0.8 mmol/L (https://diagnostics.roche.com/global/en/products/instruments/accutrend-plus.html#productSpecs). Therefore, to run statistical analysis, we decided to adopt the value of 0.7 in the measurements that showed the message 'LOW', even knowing that the real value could be lower than this. Although this limitation, we understand the lactate serum data is quite important to show the reader the capture was fast and not stressful.

- line 120: Why didn't you use the measured body temperature? 122 pH2O=47 mmHg - please justify this assumption. It seems to me that you assume that the temperature of the expired air has temperature of 37°C and saturated water vapour content at that temperature. I doubt this is correct. The maxilloturbinals in the nasal cavity will for sure preserve some heat and water, resulting in lower expired air temperature and lower water vapor pressure in the expired air.

Answer: The capybaras’ body temperature was measured by rectal route (line 132-133 and Table 1) but it was assumed 37°C for [P(A-a)O2] calculation, since it is not possible to obtain the real temperature in the alveoli. This methodology has been used in other studies as Fahlman Å, Woodbury M, Duke-novakovski T, Wourms V. Low flow oxygen therapy from a portable oxygen concentrator or an oxygen cylinder effectively treats hypoxemia in anesthetized white-tailed deer (Odocoileus virginianus). J Zoo Wildl Med. 2014;45(2):272–7

- line120: RQ=1 - Why? Normally assumed RQ is 0.85.

Answer: Thank you for the question. RQ = 1 was adopted because it concerns values of a strictly herbivorous species.

- Also please explain all abbreviations. PAO2 and PiO2 are not explained.

Answer: We apologize about that. All of the abbreviations have been explained in this revised version.

- line 136: Why did you use one-tailed paired t-test? For single tailed test you need to state the hypothesis first (greater/lower). 

- lines 149-150: missing Test statistics, p levels, df. Pleas provide this data and the tests for the revision process or even better place them in an appendix.

Answer: Significant p levels were added in this revised version. We used one-tail because we increased the O2 flow so, we would expect only an increase in the PaO2. In this way, a one-tail test is the best choice to do. An important Reviewer’s remark was about paired or unpaired data. Indeed, the corrected test is unpaired, which we have already applied in this paper. Thank you very much for this remark. This issue occurred because we did statistical analysis considering three groups at 30 min in the draft version. However, we understood all of the capybaras were given the same condition at 30 min and then, we decided to put them all together at that moment. We re-run the statistical analysis, applying the “unpaired” test now. We apologise for this mistake.

- line 136: What do you mean by intragroup?

Answer: This part was rewritten in this revised version (lines 157-161).

- line 146: "The capybaras remained quiet after they were caught in the trap." How do you know this? Were the traps observed?

Answer: Yes, we followed animals behaviour throughout the procedure.

- line 146: "The small size of the traps facilitated physical restraint with the net for drug administration by hand injection." This belongs to methods.

Answer: Thank you for the comment. This part was removed from the revised version.

- line 148: not meaningful to pool the two protocols at this moment. You explain the pooling below. Please check the mean, my calculations show 9.9min mean induction time.

Answer: Sorry about that. This part was rewritten in this revised version, pointing out the induction times from the DMM and DMB protocols (lines 167-169).

- line 153: Please define hypoxaemia in methods. What is the threshold for hypoxaemia?

Answer: Thank you for this remark. Hypoxaemia definition was added in the revised version (line 110-111).

- line 154: This information is not shown in Table 1. Only means are presented in Table 1. Please reformulate the sentence. Check/remove the ">" sign.

- line 157: difference in what?

Answer: This part was written in this revised version (lines 177-179).

- line 160: Please list all abbreviation in the table caption. Table should be able to stand alone. Valid for Tab. 2 as well.

- Table 2: Why is this table separate from table 1? Consider merging the tables.

Answer: Thank you for this suggestion. We have made them, as suggested.

- Tab 1 3rd column. Group does not seem to be a good name. Moreover this column is the same in whole table and therefore redundant. Please remove. Same for table 2

Answer: We changed “sub-group” for “treatments”. This information was added in this revised version.

- SaO2: It seems that you compared percentages by parametric test without any transformation. Please justify, the usual approach is to perform arcsine transformation prior to statistical comparison.

Answer: The SaO2 was considered parametric data, according to the statistical analysis.

- PaO2/FiO2 please add to methods how was this calculated. Add a unit. And consider removing from the table as it is not used in the second part of the table.

- P(A-a)O2 Consider removing from the table or filling the other part of the table.

175 this wording suggest gradual increase, but only two measurements do not allow this statement.

Answer: Thank you for the question. We have made them, as suggested.

- Unify the units throughout the text. Please express the units per Liter or justify. I also suggest to use SI units, ie mol/L everywhere where possible.

Answer: Thank you for the question. We have made them as suggested.

- Lactate: please check if median really equals lower boundary of interquartile range. The raw data suggest median of 1 and not 0.7 in the 30 min group. Please check the other values as well. Moreover the numbers do not seem to be correct because of the assumption on line 116.

Answer: Thank you for this remark. The serum lactate median (0.75 mmol/L) was added in the new version of the raw data and was corrected in this revised version (Table 1). We understand the other comment has been answered aforementioned.

- Na+ and K+ data missing for one animal in the raw data and the methods state that animals with missing data were excluded. Please adjust.

Answer: Thank you for this remark. This information has been added in this revised version (raw data). 

- line 200 Fig 1. Consider erasing the figure and putting the correlations in the table1/2. "BE" not explained anywhere in the text, please delete it from the figure. In the image legend on the right change the decimal comma to decimal point. Remove the legend bellow the figure and put the text in figure caption.

Answer: Thank you for this suggestion. We have made them as suggested.

- line 204: Was the p-value adjusted for multiple tests? if yes, please add this information to methods. If not, consider doing so. With 22 test the probability of false positive is very high.

Answer: No, it was not. 

- line 222: patients?

Answer: Sorry about that. The text has been changed in this revised version (line 257).

- line 226: But how reliable and comparable with reality these figures are?

Answer: Although the calculated indices use reference values and not the actual value, the data indicate that an impairment of oxygen exchange may have occurred. Thus, it might explain the low PaO2 values observed at 30 minutes. Probably due to hypoventilation, as neither altitude nor barometric pressure was high. There was no reason to suspect intrapulmonary causes, as the animals were clinically well before and after capture.

- lines 228-230: Ratios do not have dimension. Additionally the sentence does not have verb and is meaningless, please reformulate.

Answer: Thank you for this suggestion. This part was rewritten in this revised version (line 264-265). 

- line 234: It seems to me that PaO2 increased in ALL animals in 2L group. Moreover I suggest not to mix PaO2 and hypoxaemic. Please be careful in formulations.

Answer: Thank you for this remark. This information was corrected in this revised version (lines 270-271).

- line 239: "animals that were breathing air in the 2L group" is not clear. Two meanings are possible. Either delete the breathing air or 2L group.

Answer: This part was rewritten in this revised version (line 276-277). 

- 259 why however?

Answer: Sorry about that. This part was rewritten in this revised version (lines 298-299).

- line 260: please check the units. You are using mmol/L here, but in table 2 you are using mg/dL with the same parameter. One of these is probably faulty.

Answer: Lactate plasma concentration was measured as mmol/L, indeed. This information was corrected throughout the text.

- line 265: If no long-term adverse effect of the anesthesia on the animals was observed, how necessary is to prevent hypoxaemia then?

Answer: Thank you for this remark. The text was rewritten in this revised version (line 304).

---

## [Decision Letter · Decision Letter 1]

10 Nov 2021

Intranasal oxygen reverses hypoxaemia in immobilised free-ranging capybaras (Hydrochoerus hydrochaeris)

PONE-D-21-16924R1

Dear Dr. Carregaro,

We’re pleased to inform you that your manuscript has been judged scientifically suitable for publication and will be formally accepted for publication once it meets all outstanding technical requirements.

Kind regards,

Ruud AW Veldhuizen

Academic Editor

PLOS ONE

Additional Editor Comments (optional):

Reviewers' comments:

Reviewer's Responses to Questions

**Comments to the Author**

1. If the authors have adequately addressed your comments raised in a previous round of review and you feel that this manuscript is now acceptable for publication, you may indicate that here to bypass the “Comments to the Author” section, enter your conflict of interest statement in the “Confidential to Editor” section, and submit your "Accept" recommendation.

Reviewer #1: All comments have been addressed

2. Is the manuscript technically sound, and do the data support the conclusions?

Reviewer #1: Yes

3. Has the statistical analysis been performed appropriately and rigorously? 

Reviewer #1: I Don't Know

4. Have the authors made all data underlying the findings in their manuscript fully available?

Reviewer #1: Yes

5. Is the manuscript presented in an intelligible fashion and written in standard English?

Reviewer #1: Yes

6. Review Comments to the Author

Reviewer #1: (No Response)

7. PLOS authors have the option to publish the peer review history of their article (what does this mean?). If published, this will include your full peer review and any attached files.

Reviewer #1: No

---

## [Editor Report · Acceptance letter]

18 Nov 2021

PONE-D-21-16924R1 

Intranasal oxygen reverses hypoxaemia in immobilised free-ranging capybaras (*Hydrochoerus hydrochaeris*) 

Dear Dr. Carregaro:

I'm pleased to inform you that your manuscript has been deemed suitable for publication in PLOS ONE. Congratulations! Your manuscript is now with our production department. 

Kind regards, 

on behalf of

Dr. Ruud AW Veldhuizen 

Academic Editor

PLOS ONE